# Optical Flow-Based Full-Field Quantitative Blood-Flow Velocimetry Using Temporal Direction Filtering and Peak Interpolation

**DOI:** 10.3390/ijms241512048

**Published:** 2023-07-27

**Authors:** Liangwei Meng, Mange Huang, Shijie Feng, Yiqian Wang, Jinling Lu, Pengcheng Li

**Affiliations:** 1Britton Chance Center for Biomedical Photonics and MoE Key Laboratory for Biomedical Photonics, Wuhan National Laboratory for Optoelectronics, Huazhong University of Science and Technology, Wuhan 430074, China; marwen@hust.edu.cn (L.M.); huangmange@hust.edu.cn (M.H.); m202173502@hust.edu.cn (Y.W.); jinglinglu@mail.hust.edu.cn (J.L.); 2Research Unit of Multimodal Cross Scale Neural Signal Detection and Imaging, Chinese Academy of Medical Science, HUST-Suzhou Institute for Brainsmatics, Jiangsu Industrial Technology Reserch Institute (JITRI), Suzhou 215100, China; 3Department of Biomedical Engineering, Hainan University, Haikou 570228, China

**Keywords:** optical flow, direction filtering, temporal peak interpolation, quantitative, blood flow

## Abstract

The quantitative measurement of the microvascular blood-flow velocity is critical to the early diagnosis of microvascular dysfunction, yet there are several challenges with the current quantitative flow velocity imaging techniques for the microvasculature. Optical flow analysis allows for the quantitative imaging of the blood-flow velocity with a high spatial resolution, using the variation in pixel brightness between consecutive frames to trace the motion of red blood cells. However, the traditional optical flow algorithm usually suffers from strong noise from the background tissue, and a significant underestimation of the blood-flow speed in blood vessels, due to the errors in detecting the feature points in optical images. Here, we propose a temporal direction filtering and peak interpolation optical flow method (TPIOF) to suppress the background noise, and improve the accuracy of the blood-flow velocity estimation. In vitro phantom experiments and in vivo animal experiments were performed to validate the improvements in our new method.

## 1. Introduction

Microvessels play an important role in regulating hemodynamics in the body, and respond early in many diseases. The accurate measurement of microvascular blood-flow velocities often allows the early detection of diseases associated with microvascular disease. Over the years, increasing efforts have been devoted to pursuing a high-resolution quantitative blood-flow velocimetry, to understand the mechanism of pathology in diseases [1,2,3]. Doppler-based velocimetry methods, such as laser Doppler imaging (LDI) [4], photoacoustic Doppler velocimetry (PDV) [5], and Doppler optical coherence tomography (DOCT) [6,7,8] obtain the blood-flow velocity derived from the frequency shift (defined by the formula f=(v/λ)cosθ). These techniques depend on the Doppler angle θ, so are usually not sensitive to the blood flow perpendicular to the detecting beam. Laser speckle contrast imaging (LSCI) [9,10,11,12,13,14] maps the two-dimensional blood-flow speed at a high spatiotemporal resolution, by estimating the decorrelation time (τ_c_) of the electric field of the dynamic speckle statistics. Dynamic light scattering imaging (DLSI) [15] and diffuse correlation spectroscopy (DCS) [16,17] fit the temporal changes in the light intensity with theoretical light intensity autocorrelation functions, to obtain the decorrelation time of the electric field. However, both LSCI and DLSI present the challenges of measuring the absolute speed of moving blood cells [18,19]. To obtain the absolute velocity of blood flow, several particle image velocimetry (PIV) algorithms [20,21,22] have been developed to track the movement of blood cells, in consecutive reflectance or fluorescence image frames recorded by various imaging modalities, such as wide-field microscopy [23,24,25], confocal microscopy [26,27], two photon microscopy [28,29], and optical coherence tomography [30]. Correlation PIV calculates the spatial cross-correlation of the light intensity between consecutive frames within a local window, to obtain the speed and the direction of the moving red blood cells. However, the resolution and accuracy of correlation-based PIV will be affected by the size of the interrogation window, and the number and uniformity of the tracer particles in the window [31]. The velocity of moving blood cells can also be measured by estimating the slope of the spacetime diagram of the light intensity changes in a line selected along a blood vessel [32]; however, this also results in the sacrifice of spatial resolution. To improve the spatial resolution of PIV, optical flow analysis is introduced, to trace the blood-flow velocity [33,34,35,36,37,38]. Given the assumption of constancy and similarity in the local brightness patterns in classic optical flow theory, the variation in the pixel brightness between consecutive frames is used to calculate the direction and speed of the moving blood cells. Compared with correlation PIV, optical flow analysis not only increases the spatial resolution, but also the upper limit of the measurement of blood-flow speed [39,40]. However, the use of the traditional optical flow algorithm to calculate the blood-flow velocity faces the challenges of strong noise from the background tissue, and a significant underestimation of the blood-flow speed in big blood vessels, due to errors in detecting feature points in optical images.

In this paper, we propose an optical-flow-based full-field quantitative blood-flow velocimetry, using temporal direction filtering and peak interpolation to suppress background noise and improve the accuracy of the blood-flow velocity measurement. The results of our in vitro phantom experiments, and in vivo animal experiments demonstrate well the performance of our new method.

## 2. Results

### 2.1. In-Vitro Phantom Experiment of RBC Flow in Glass Capillaries

The results of the in vitro simulation experiment are shown in Figure 1B, with a schematic of the simulation experiment displayed on the left, and a scatter plot of the preset flow rate, compared to the TPIOF-measured velocity, displayed on the right. The obtained *R*^2^ value of 0.991, which matches the speed estimated by TPIOF at the preset flow rate, indicates a significant linear correlation between the estimated speed and the preset flow rate.

### 2.2. In Vivo Blood-Flow Estimation

After being validated through an in vitro experiment, the accuracy of TPIOF was further verified using an in vivo experiment. A set of consecutive image sequences is computed by using the HSOF and TPIOF methods, then V_HSOF_(i) and V_TPIOF_(i) were obtained (as shown in Figure 2B). Two regions of interest (ROIs), V1 (artery) and V2 (vein), were selected on the V_HSOF_(i) and V_TPIOF_(i) maps, to measure the temporal variation in the flow speed. The results are presented in Figure 2C. The blue curve represents the speed variation estimated by TPIOF, while the light purple curve corresponds to HSOF. The complex movement of red blood cells (RBCs) within the vasculature results in transient changes in the image brightness patterns. This led to a significant underestimation in the HSOF assessment, as shown by the red area in Figure 2C. It is evident from Figure 2C that the TPIOF results of arterial blood-flow waveform morphology are closer to the pulse wave than the HSOF results. The cross-sectional curves of the vessels (vessels 1 and 2, marked in white in Figure 2B) are shown in Figure 2D, to demonstrate the benefits of TPIOF in terms of the signal-to-noise ratio and vessel discrimination. The figure clearly demonstrates the smoother curve of TPIOF compared to HSOF. Notably, on the HSOF flow map, at the location of vessel 2 (capillaries), the blood vessels have been completely submerged in the background noise, while TPIOF can clearly depict the vessels. In order to evaluate the capability of TPIOF in assessing the full-field blood flow, data from the middle cerebral arterial vascular region (Figure 2E) were collected over 200 consecutive frames, and the computed results are presented in Figure 2F. The results demonstrate that TPIOF effectively captures a wide dynamic range of blood-flow values, spanning from the M4 Segment of the MCA to the capillaries. To provide a comprehensive visualization of the blood-flow values in all vessels, the color mapping in the blood-flow pseudo-color map represents the logarithmic index (log10) of the flow-rate values.

In addition, the accuracy of the measurement of the blood flow in vivo using TPIOF was verified by comparing the consistency of the measurements with those of the spacetime image velocimetry (as shown in Figure 3). Figure 3B illustrates the temporal variation in the speed at V_1_ and V_2_ in Figure 3A (top left). The TPIOF (blue) and spacetime speed curves (orange) are depicted, respectively. The results demonstrate a good correlation between the two curves. Subsequently, the velocities were measured at 48 vessels using the TPIOF, HSOF, and spacetime methods. The results of TPIOF and HSOF were compared with the spacetime results by creating scatter plots (Figure 3D). Here, the horizontal coordinates represent the evaluated speeds obtained using the spacetime method, while the vertical coordinates represent the speeds evaluated using TPIOF and HSOF. The results of TPIOF vs. HSOF show that the slope of linear fit of TPIOF is closer to 1 (0.944 vs. 0.760), and the R^2^ value is better (0.94 vs. 0.91), which implies that TPIOF has a better accuracy.

Figure 3C shows the results of the flow-rate calculations for the vascular branches (e.g., Figure 3A, lower left) V3_1, V3_2, and V3_3. The histogram indicates no significant difference in the sum of the flow rate of V_1_1_, V_1_2_, and V_1_3_, validating the conservation of the flow rate. Figure 3E displays the velocity direction map (left), and the flow velocity intensity map (right) obtained using both the HSOF and TPIOF methods. Compared to HSOF, TPIOF exhibits a more uniform blood-flow distribution in the vascular region, and lower noise in the non-vascular region, indicating a better signal-to-noise ratio, which is also evidenced by the results of the time-domain curve comparison in Figure 2D. Figure 3F illustrates the variation in the velocity vectors with time for the positions P1 and P2 in Figure 3E. It is obviously demonstrated that TPIOF has a better signal-to-noise ratio and temporal correlation.

### 2.3. Comparative Experiments with Correlation PIV

To validate the potential of correlation PIV in measuring the blood-flow velocity under low coherent light sources, we utilized both the correlation PIV and TPIOF methods to evaluate the same dataset, and the results are presented in Figure 4A. By comparing the results of the TPIOF and correlation PIV methods, it is evident that the TPIOF method outperforms correlation PIV in terms of the velocity assessment accuracy, and the resolution of vascular structures. The TPIOF method’s advantage in identifying smaller vortex details is demonstrated trough the comparison of results presented in Figure 4B. Overall, the TPIOF method exhibits a superior spatial resolution when compared to correlation PIV (as shown in the black dashed box in Figure 4B), thereby making it more suitable for wide-field blood-flow velocity assessment under low coherent light illumination conditions. TPIOF is a dense optical flow; every pixel point is involved in the velocity calculation. As there are no bright particles in the background region, the velocity information cannot be obtained, resulting in an uneven velocity distribution in the region of large vortices.

## 3. Discussion

This study presents a novel quantitative imaging technique, TPIOF, to assess the quantitative assessment of blood-flow velocity. The traditional HSOF method estimates the motion field by analyzing the brightness variation in successive image frames, based on the assumptions of brightness constancy and small motion. This method has been employed to estimate blood-flow velocity [33,41,42,43,44,45]. However, it faces several challenges in in vivo blood-flow velocity estimation.

Firstly, when extracting the motion of red blood cells, the accuracy of optical flow methods is often disturbed, due to the strong background noise. To address this issue, the temporal direction filtering algorithm can effectively suppress background noise. On the other hand, the rapid motion of red blood cells within blood vessels can lead to motion blurring and tracking loss. This may be a major contributor to the underestimation of speed in the optical flow method. To resolve this problem, the TPIOF method recovers the lost speed by extracting the speed peaks from the variation in the speed at the corresponding position of the image in the time series, then interpolating these peak points using quadratic polynomials. In vitro and in vivo experiments demonstrated the accuracy of this method. In addition, increasing the acquisition frame rate contributed to reducing the effect of fast motion. In this study, the maximum acquisition frame rate of the camera was extended from 110 fps to 2200 fps by reducing the active pixels of the camera to 2048 × 48. Subsequently, through setting the ROI offset, the full-frame flow-speed information was obtained. This method greatly extends the velocity estimation range of TPIOF, without the loss of spatial information. Finally, artifacts of particle motion during the exposure time reduce the accuracy of the feature detection, posing a challenge to the accurate tracking of erythrocyte motion using optical flow methods. To overcome this problem, a low-coherence green light source for pulsed illumination, with a duration of 45 microseconds (us), is employed, as a replacement for the commonly used halogen lamp illumination. This approach significantly enhances the image contrast, and reduces motion artifacts during the exposure time.

In this study, the optical system was only suitable for acquiring two-dimensional images at depths within 100 um, and could not perform three-dimensional velocity correction. Velocity correction in three dimensions would make for an interesting work. In the future, 3D structural maps of blood vessels could be obtained by two-photon and other scanning techniques, meaning that z-direction angles could be obtained for 3D velocity correction. In the future, this could be validated by analyzing phantom experimental data from 3D-printed simulated blood vessels [46,47].

Although the TPIOF method realized the in vivo blood-flow quantitative estimation, some challenges remain to be addressed. In this study, a green light source (532 nm) is utilized to enhance the image contrast in the small-vessel region, which leads to a rather weak signal reflected by the large-diameter-vessel region, limiting the measurement range of TPIOF. To address this problem, a multispectral illumination, combined with TPIOF, is helpful, to select different illumination wavelengths based on the spectral absorption properties of red blood cells, and fuse the results. This improves both the resolution of the capillaries, and the signal-to-noise ratio in large vessel regions.

To further expand the measurement range of TPIOF for blood-flow velocity, cameras that allow for a higher frame rate can be considered. Postnov et al. employed a camera with the exceptionally high frame rate [15] of up to 22,881 fps for ROI (i.e., 1280 × 32 pixels). This high frame rate enables the theoretical measurement of blood-flow velocities up to 60 mm/s, rendering it suitable for studying arterial blood rheology diseases [48]. However, in practice, to ensure a proper signal-to-noise ratio, the illumination power and camera frame rate need to be controlled within the tissue safety limit [49]. Additionally, the resolution of image acquisition systems [50,51] and the stability of optical flow methods [52,53,54] can be enhanced, to extend the range of velocity measurement, and improve the accuracy of optical flow methods.

The TPIOF method, with its ability to quantitative blood velocity measurements and discriminate capillary structures, is expected to boost the study of molecular regulatory mechanisms of post-injury traumatic vascular functional remodeling. For example, neutrophil extracellular traps (NETs) released by neutrophils impair revascularization and vascular remodeling after stroke were investigated in a study by Lijing Kang et al. [55]. Joachim Pircher [56] proposed that the cathelicidin LL-37/CRAMP plays a significant role in platelet activation and thrombo-inflammation. Qingqing Yu’s [57] study showed that AMPK activation by ozone therapy inhibits tissue factor triggered intestinal ischemia and ameliorates chemotherapeutic enteritis. A study by Lu et al. [58]. found that growth differentiation factor 11 (GDF11) promoted neurovascular recovery after stroke in mice. In addition, Rong Zhao’s [59] study showed that Cathepsin K (Ctsk) knockout exacerbated haemorrhagic transformation induced by recombinant tissue plasminogen activator after focal cerebral ischaemia in mice. In these studies, changes in blood flow velocity serve as the important intuitive feedback indicators of molecular regulatory processes, and researchers use multiphoton microscopy or confocal microscopy to reflect blood flow movement by monitoring the movement of fluorescent markers injected intravascularly. However, TPIOF tracks blood flow velocity without the need for fluorescent markers. Hence, the TPIOF is suitable for assisting in molecular regulatory studies. Additionally, tracking the trajectory and velocity of fluorescent probes using the TPIOF method will help to deepen the understanding of the process of molecular regulatory mechanisms [60].

## 4. Materials and Methods

### 4.1. Optical Imaging System

Figure 5A illustrates the setup of the TPIOF image acquisition system. In order to minimize the motion artifacts of RBCs, a signal generator (Uni-Trend, Dongguan, China) was employed to output a pulse signal, which enabled synchronized green LED illumination (Cree, Durham, NC, USA) and image triggering (Basler, Ahrensburg, Germany), as shown in the timing diagram in Figure 5A (upper). The square wave duty cycle of the LED driver was set to 45 us, and connected to a custom-made LED driver module. This module regulated the illumination time of the light source by controlling the pulse width. The reflected photons were collected using a zoom microscope (Simopto, Wuhan, China) with an optical magnification of 3.5X. To overcome the limitations imposed by the camera’s transmission speed, the maximum acquisition frame rate of the camera was extended from 110 fps to 2200 fps by configuring the active pixels to 2048 × 48, with a camera exposure time of 400 µs. Two hundred images were recorded, to obtain the blood-flow velocity using TPIOF. A full frame of blood-flow velocity (2048 × 1536) was obtained by adjusting the position of the active pixel, and performing 32 scans. This approach significantly reduced the dependence on high-speed cameras. The averaged TPIOF results are presented in Figure 5C, where the colors indicate the flow direction, and the color contrast represents the normalized velocity magnitude.

### 4.2. Temporal Direction Filtering and Peak Interpolation Optical Flow

Figure 2A illustrates the workflow of TPIOF. The Horn–Schunck optical flow algorithm (HSOF) [61], which utilizes image intensity gradients, is applied to a sequence of successive images *I*_1_, *I*_2_, …, *I_n_*, *I*_*n*+__1_ captured by a high-speed camera, to calculate the velocity components in the horizontal (*u*) and vertical (*v*) directions. The principle of HSOF can be briefly described as follows.

Firstly, we define *I*_1_(*x*, *y*, *t*) as the initial image intensity, where *x* and *y* represent the pixel horizontal and vertical location in the image coordinates, and t denotes time. Let *I*_2_(*x*, *y*, *t*) represent the pixel intensity after a short time duration *dt*, with *dx* and *dy* as the pixel displacements in the image. The fundamental assumption of an optical-flow-based method is that the gray scale of the image of the moving particles remains constant over short intervals. This constancy can be described mathematically as follows:(1)$$I_{1}(x,y,t)=I_{2}(x+dx,y+dy,t+dt)$$

By performing a first-order Taylor expansion on *I_2_*, with velocity *u*(*x*,*y*) = *dx*/*dt*, *v*(*x*,*y*) = *dy*/*dt*, and denoting the partial derivatives of the image intensity as *I_x_*, *I_y_*, and *I_t_*, we have the linearized version of the intensity constancy assumption:(2)$$I_{x}u(x,y)+I_{y}v(x,y)+I_{t}=0$$

Equation (2) is the classical optical-flow [61] equation. Since *u* and *v* are two unknowns, this equation system is underdetermined, which also constitutes the well-known aperture problem. To solve this problem, HSOF introduces an additional assumption: the global smoothness constraint condition of optical flow; i.e., the change of *u* and *v* with the movement of the pixel points is slow. That speed-smoothing term can be described mathematically as follows:(3)$$\zeta_{c}^{2}={\left(\frac{\partial u}{\partial x}\right)}^{2}+{\left(\frac{\partial u}{\partial y}\right)}^{2}+{\left(\frac{\partial v}{\partial x}\right)}^{2}+{\left(\frac{\partial v}{\partial y}\right)}^{2}$$

The first assumption allows us to obtain Equation (2); however, the unique solution for the horizontal velocity component *u* and the vertical velocity component *v* cannot be determined by solving Equation (2) alone (i.e., there are multiple uncertain solutions for *u* and *v*). In order to determine the unique velocity solution, the second assumption, i.e., Equation (3), needs to be associated as a constraint to obtain the unique solution, so that the optimal velocity solution can be obtained by calculating the minimum of Equation (4).
(4)$$L={\left(I_{x}u+I_{y}v+I_{t}\right)}^{2}+\alpha^{2}\left[{\left(\frac{\partial u}{\partial x}\right)}^{2}+{\left(\frac{\partial u}{\partial y}\right)}^{2}+{\left(\frac{\partial v}{\partial x}\right)}^{2}+{\left(\frac{\partial v}{\partial y}\right)}^{2}\right]\to \min$$
where α is the smoothing weight coefficient, indicating the weight of the velocity-smoothing term. Then, *u*(*x*,*y*) and *v*(*x*,*y*) can be obtained by iteratively solving the minimum value of Equation (4). This equation can be solved using the Euler–Lagrange equation; the solution process has been described in detail in the literature. Then, the HSOF velocity map *V*(*x*,*y*) can be obtained by:(5)$$V(x,y)=\sqrt{u{\left(x,y\right)}^{2}+v{\left(x,y\right)}^{2}}$$

In comparison to HSOF, TPIOF offers two distinct advantages. Firstly, it addresses the issue of underestimated velocity. Secondly, it enhances the visibility and differentiation of blood vessels. The underlying principle of this method can be elucidated as follows.

Firstly, we use the HSOF method to calculate the blood-flow velocity vector *V_HOSF_*(*x*,*y*) for each frame in a set of data. We define the angles between *V_HOSF_*(*x*,*y*) and its horizontal velocity component *u_i_*(*x*,*y*) and vertical velocity component *v_i_*(*x*,*y*) as *α*(*i*) and *β*(*i*), respectively, which can be calculated using the inverse tangent function:(6)$$\alpha (i)=arctan\frac{v_{i}(x,y)}{u_{i}(x,y)};\beta (i)=arctan\frac{u_{i}(x,y)}{v_{i}(x,y)}$$

Then, we calculate the average angle *θ* at each pixel position of the *M* velocity images, which is defined as:(7)$$\theta =\frac{1}{M}\left(\sum_{i=1}^{M}\alpha_{i}+\sum_{i=1}^{M}\beta_{i}\right)$$

The noise introduced during image acquisition is also recognized by the optical flow as a feature point, so the angle *θ* of the velocity vector computed by the optical flow result contains the signal component *θ_signal_*, and the noise component *θ_noise_*; i.e., *θ* = *θ_signal_* + *θ_noise_*. The experiments found that the noise obeys a symmetric distribution (refer to Figure 5A upper right); i.e., <*θ_noise_*> ≈ 0. Therefore:(8)$$V_{\mathrm{out}\;}=V_{i}\times |\sin\theta |\approx V_{i}\times |\sin\theta_{signal}|$$

After applying temporal directional filtering, the signal peaks of each pixel on the image over the time series are extracted using the ‘findpeaks’ function in MATLAB. To eliminate high-frequency interference, a minimum peak width of 3 (as set in this study) is employed. Subsequently, the extracted peak points are connected through quadratic polynomial interpolation, to generate the final output signal methods V_TPIOF_.

### 4.3. In Vitro Experiment

To evaluate the accuracy of TPIOF for estimating the flow velocity of red blood cells (RBCs), a phantom experiment was performed. Fresh blood samples were collected from mouse tails, using a vacuum 2 mL EDTA K2 vacuum blood collection tube (Hangzhou Ciping Medical Equipment Co., Ltd., China). The blood was then mixed with PBS at a ratio of 1:10. Subsequently, the diluted blood sample was injected into a glass capillary with a diameter of 300 μm, using a 30 μL microinjection syringe pump (Shanghai Gaoge Microsyringe, China) (Figure 1A). To simulate the blood-flow patterns at different flow rates, the diluted blood samples were propelled using a micropump (TJ-4A, Baoding Langhe Constant Flow Pump Co., Ltd., China). The injection rate was incremented by 2 μL/min within the range of 2 to 20 μL/min. A total of 10 datasets were collected, with each set repeated five times, to ensure data reliability. The collected data were processed using TPIOF, and the results are presented (Figure 1B).

### 4.4. In Vivo Experiments

Adult male C57 mice weighing approximately 30 ± 2 g were chosen as the experimental subjects. Anesthesia was induced in the mice via the intraperitoneal injection of a mixture containing 10% urethane and 2% chloral hydrate (100 mL/kg). After the completion of anesthesia, the scalp was incised to expose the skull. A flat region between the lambda and bregma was chosen, and a 0.8 mm diameter dental drill (RWD 78001) was utilized to make a 3 to 4 mm diameter craniotomy. Subsequently, a piece of 4 mm diameter coverglass was positioned over the craniotomy, and affixed with cyanoacrylate adhesive (Vetbond, 3M), for stabilization. Dental cement (Super Bond C&B) was used to fill the exposed area of the skull, ensuring a sustained intracranial pressure. A heating pad was employed throughout the surgical procedure, to maintain the mice’s body temperature at 37 °C, ensuring experimental stability. Our previous experience indicated a relatively lower rate of regeneration in the meningeal blood vessels and skull bone after the cranial window creation, maintaining an extended period of window clarity.

Additionally, the flow conservation at vessel V_3_ (Figure 3A, bottom-left) was verified through the comparison of the flow rates of V_3-1_ and the sum of V_3-2_ and V_3-3_, as depicted in Figure 3C. To minimize measurement errors, ten sets of cross-sectional flow rates were obtained at various locations along the three vessels, and the statistical results were visualized using histograms.

### 4.5. Comparative Experiments with Correlation PIV

To illustrate the superior spatial resolution of TPIOF, we acquired a standard dataset from the PIV Challenge website (www.pivchallenge.org/, accessed on 12 August 2022), and conducted a comparison using the lower-left section of the dataset. This specific portion showcases a random two-dimensional sinusoidal vortex flow. To ensure a consistent computation time, we applied an 8*8 interrogation window, and a 4-pixel step for the correlation PIV method. The results of the associated PIV and TIPOF were then put into Figure 4B, along with the ground truth, for comparison.

### 4.6. Spacetime Image Velocimetry

The accuracy of the TPIOF measurement of in vivo blood flow is verified by comparing the consistency of the results of the spacetime image velocimetry method. In the space-time velocimetry method, a segment of a line along the center of a blood vessel is selected on an image, and the intensity values of the consecutive frames of the line pixels are extracted, to put together a two-dimensional picture, as shown in Figure 3A (left). The high-contrast stripes of the image can be clearly seen, and the angle of the stripes is calculated using the Radon transform [41,42]. As the time intervals and the corresponding spatial distances of neighboring pixels are known, the centerline velocity of the blood flow can be calculated by solving the inverse cotangent of the angle of the stripes.

## 5. Conclusions

We proposed the TPIOF method to quantitatively estimate full-field blood-flow velocity. This method addresses issues commonly found in conventional optical flow methods, such as background noise and velocity underestimation, through the implementation of temporal direction filtering, and peak interpolation algorithms. The TPIOF method facilitates the acquisition of the quantitative blood-flow velocity and vascular structure of different vessel sizes under non-contact conditions, without the need for contrast agents. In particular, it can clearly distinguish capillary structures. The reliability of the TPIOF method has been demonstrated through in vitro phantom experiments, and quantitative assessment experiments of in vivo cerebral blood flow in mice. In the future, utilizing quantitative measurements, we could make a significant contribution to the fundamental research on cardiovascular diseases, and the early diagnosis of clinical microcirculation diseases.

## Figures and Tables

**Figure 1 ijms-24-12048-f001:**
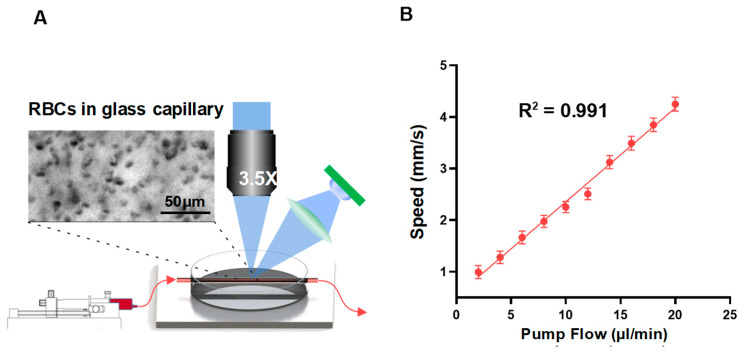
The results of an in vitro phantom experiment are presented. Diluted blood samples were flowed through glass capillaries at a controlled flow rate, using a microinjection syringe pump. (**A**) The flow speed was evaluated using the TPIOF method, through a sequence of images acquired with a 3.5X objective lens. (**B**) The relationship between the measured speed using TPIOF, and the preset flow rate of the microinjection syringe pump, ranging from 2 to 20 µL/min, was examined over 10 trials (i.e., n = 10).

**Figure 2 ijms-24-12048-f002:**
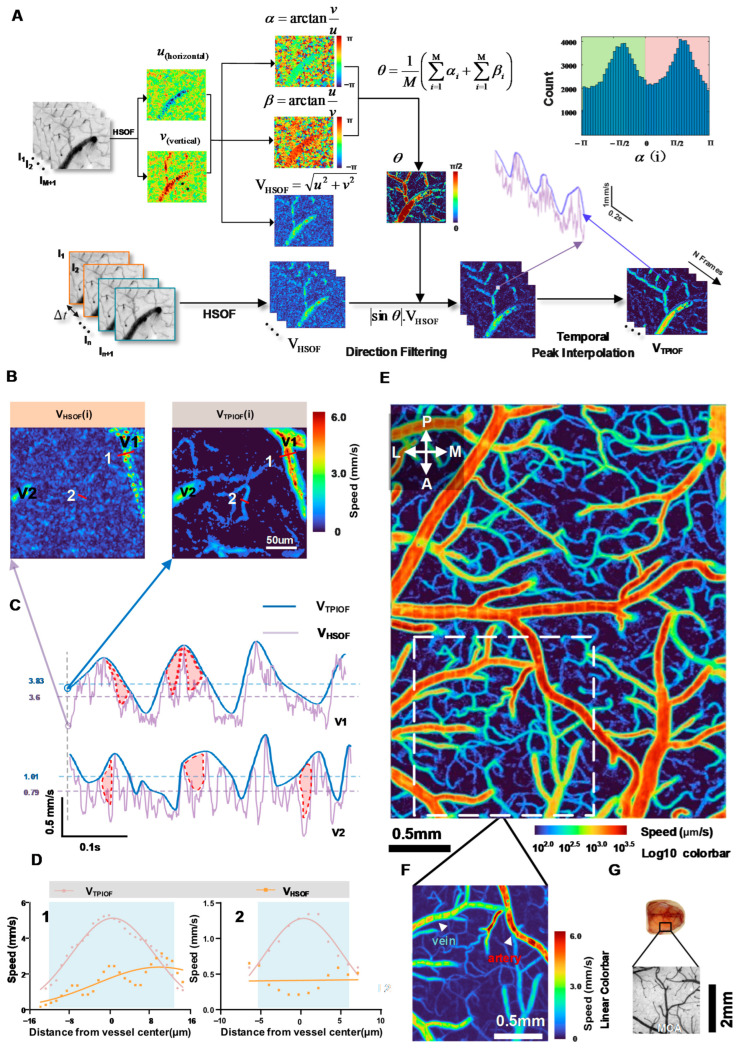
The algorithm flowchart of the TPIOF and in vivo experiment results. (**A**) The flowchart of the TPIOF method, and the histogram of velocity angles in non-vascular regions (upper right). (**B**) Blood-flow maps of the HSOF(i) and TPIOF(i) methods. (**C**) Blood-flow speed curves over time for vessels V1 and V2 in Figure 2B (words in black); the light blue and purple dashed lines in the figure represent the average speed obtained using the TPIOF and HSOF methods, respectively. (**D**) Cross-sectional velocity profiles of vessels 1 and 2 in Figure 2B (words in white), with the vascular location indicated in light blue. (**E**) The TPIOF blood-flow map of the area indicated in Figure 1G (solid black box); the color-mapping in the blood-flow pseudo-color map represents the logarithmic index (log10) of the flow-rate values. (**F**) Linear blood-flow maps corresponding to the location of the white dashed box in Figure 1E. (**G**) Schematic representation of the head position of the mouse from which the data were collected.

**Figure 3 ijms-24-12048-f003:**
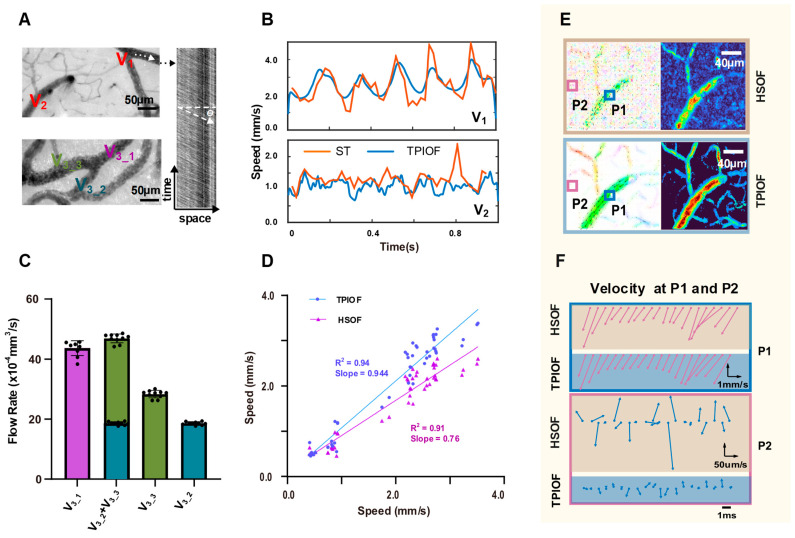
The results of the TPIOF accuracy validation experiment. (**A**) Green light reflection image (upper left), and space-time algorithm schematic (right), and the striped image represents the unfolding of pixel intensities in time along the white dashed arrows. scale 50 μm. (**B**) The temporal variation in the blood-flow speed values at positions V_1_ and V_2_, corresponding to A (upper left), space-time (orange) and TPIOF (blue). (**C**) The different colored bars correspond to the flow rate in the corresponding color-marked vessels in the lower-left panel of (**A**). (**D**) Scatterplot of the speed (*y*-axis) vs. space-time velocity (*x*-axis) for TPIOF and HSOF. (**E**) Directional color-coding velocity map (left), and speed map (right), scale 40 μm. (**F**) The temporal variation in the velocity vector at positions P1 and P2, corresponding to (**E**), with a time interval of 1 ms. Where the direction and length of the arrow represent the direction and speed of motion, respectively.

**Figure 4 ijms-24-12048-f004:**
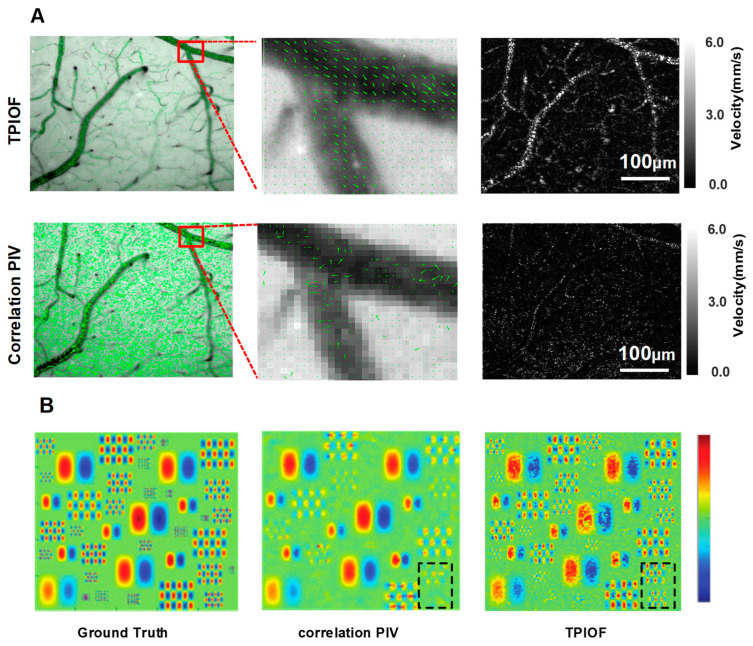
The results of the controlled validation between TPIOF and correlation PIV. (**A**) From left to right are: velocity vector map overlaid with green reflective intensity image (top-left); local magnification (top-center); and TPIOF calculation of blood flow velocity results (top-right), and the results of the correlation PIV method (bottom). (**B**) The simulated data consists of two frames containing randomly distributed vortex motions of different sizes. In order to enhance the contrast of the displayed images, only the vertical component motion of the vortex motion is shown, where red indicates upward motion blue indicates downward motion. From left to right, the true vortex map of the PIV Challenge website data, the vortex map computed using correlation-based PIV, and the vortex map computed using TPIOF, where the TPIOF spatial resolution is significantly better than the associated PIV, as can be seen from the black dashed box.

**Figure 5 ijms-24-12048-f005:**
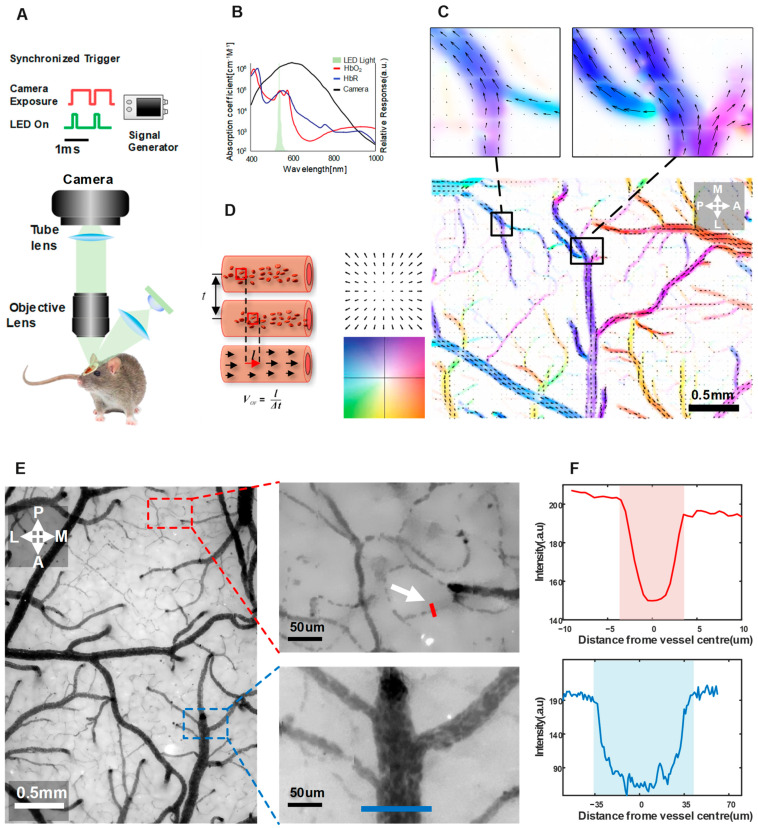
The experimental setup and presentation of results for TPIOF. The schematic of the setup involved, using a 532 nm wavelength LED as the light source to acquire high-contrast images of the blood flow. The LED beam was uniformly illuminated through a lens, which was synchronized with the camera frame-rate using a function generator for triggering. The reflected light from the sample collected by a 3.5x objective lens, then passing through a tube lens, was recorded by the cameras. (**A**) Optical imaging system diagram, which shows the optical system diagram, and a schematic of the pulsed light source and camera synchronization trigger. (**B**) The diagram displays the absorption spectra of HbR and HbO, the emission spectrum of the illumination LED, and the response spectrum of the camera. (**C**) The in vivo blood-flow velocity results, with color-coding for the optical flow velocity fields. The direction is encoded in the color hue, and the speed in the color brightness. (**D**) Schematic diagram illustrating the calculation of the optical flow velocity. (**E**) A green reflex image of the mouse cerebral cortex, and a magnified image of the localized area, the white arrows are used to highlight the location of the capillaries. (**F**) The light intensity profile along the cross-section of a blood vessel which corresponding to the location of the vessel in (**A**) (the red line corresponds to a capillary, the blue line corresponds to the vein).

## Data Availability

The data presented in this study are available upon request from the corresponding author.

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
