# Peer review of "Optical Flow-Based Full-Field Quantitative Blood-Flow Velocimetry Using Temporal Direction Filtering and Peak Interpolation"

_ijms, 2023, doi:10.3390/ijms241512048_

Round 1

Reviewer 1 Report

This manuscript is a decent one written with scientific rigor and enough significance. The introduction is good, and it has quite thorough reference list in the back. The structure is bizarre, though, as 'Materials and Methods' appears after 'Discussion'. There are many typos and grammatical error, as written below, so there are many aspects to be revised before publishing. Some questions and comments are below, followed by many minor points to be corrected.

In discussion section, authors claim that TPIOF can solve the problem of rapidly changing RBC aggregation pattern, but it is not clear how. Please elaborate on it.

In Fig 5A center column, what would be the reason that overlay image for PIV is with much lower resolution than the TPIOF counterpart?

In Fig 5B, a vortex map has been used for validation purpose. But it is not clear what this vortex map is for, as it does not bear any resemblance to any capillary network. Also, it needs to be discussed why TPIOF result shows noisy structure inside of the big vortex region, although it looks superior to PIV for small vortex region.

<Minor issues>

- Line 29 : Given the 'assumption'

- Line 71 : must be 'emition spectrum', not absorption spectrum

- Fig 1 caption : Description of (C) and (D) are mixed

- Line 140-142 : No description on Fig 4F is found

- Fig 3 E, F : the image in F is an upside-down image of the one in E. 

- Fig 3 caption : No description on E is found

- Line 149 : Where does this sentence start? Must be an error

- Fig 4 F : The scale vector for upper portion is missing

- Fig 5 caption : (top) -> (top-center)

                  (bottom) -> (bottom-center)

- Line 218 : grammatical error

- Line 254 : The acronym HSOF needs to be spelled out when the term appears first, not here.

- Line 265-266 : errors in subscripts.

- Line 286 : 'i' in vi and ui need to be subscripts

Some errors have been detected, and careful proofreading will be required.

Author Response

Dear Editors and Reviewers:
Thank you for your kind letter and suggestions regarding the submission entitled “Optical flow-based full-field quantitative blood flow velocimetry using temporal direction filtering and peaks interpolation” (Manuscript ID: ijms-2491816). We have checked and revised our manuscript carefully, and the changes have been highlighted in the revised manuscript. We hope that our effort can address the reviewers’ concerns and the revised manuscript could be satisfactory for publication in  International Journal of Molecular Sciences.
The responses to the comments please see the attachment.

Reviewer 2 Report

In the paper, authors proposed an optical flow-based full-field quantitative blood flow (BF) velocimetry with the use of temporal direction filtering and peaks interpolation to suppress the background noise and improve the accuracy of BF velocity measurement. Perhaps they came up with an interesting and new method, but the quality of the text and figures of the manuscript does not yet allow to understand how the method works and what its advantage is.

First of all, the structure of the manuscript is not standard. Мaterials and methods follow results, so the text is difficult to understand logically. Moreover, the results section, for example, page 4, also contains descriptions of materials and methods. For example: «The accuracy of TPIOF was validated using…», line 119; «… the flow conservation at vessel was verified by…», line 134, line 161-166, etc. It is necessary to formulate the purpose and objectives of the study in the introduction, and then describe in details all the materials and methods for solving problems. Then – pure results and discussion.

In general, it is possible to formulate the main questions that are not disclosed in the text and which require clarification. Only after the detailed answers to all these questions, it will be possible to judge the scientific novelty, relevance and significance. Namely:

1. Is this practical oriented or fundamental research? If practical (medical), please, describe the medical problem(s) which need the knowledge about the velocity in a separate small vessel. If fundamental, please, describe the novelty of knowledge about hemocirculation, which can be obtained by this refined method.  

2. Authors claim the noise reduction, but in section 4.2 there is no noise component of the signal and it is not clear theoretically how and what kind of noise is reduced.

3. In Eq. (4) according to the Eq. (2) the first term of the right side of the equation is equal to zero. Therefore, the description and solution is not evident. Please, clarify the situation.

4. The principle of determining an optical flow velocity as a time offset delay shown in Fig. 1D is valid only for horizontally located vessels. How is the angle of inclination of vessels measured and taken into account?

5. The authors analyze photos of vessels, but it is not clear how they see them with a microscope in vivo? What is the depth of visibility of the vessels? What is the image quality and sizing accuracy?

6.  How one can understand in the image, for example, in Fig. 3F, where are arteries and where are veins? Usually, the venous bed in the skin is several times larger than the arterial one. Therefore, in the same number of times, more vessels with venous velocity and fewer arteries with higher velocity should be visible. However, this is not visible in Fig.3F. How can authors explain this? 

Author Response

Dear Editors and Reviewers:

Thank you for your kind letter and suggestions regarding the submission entitled “Optical flow-based full-field quantitative blood flow velocimetry using temporal direction filtering and peaks interpolation (Manuscript ID: ijms-2491816). We have checked and revised our manuscript carefully, and the changes have been highlighted in the revised manuscript. We hope that our effort can address the reviewers’ concerns and the revised manuscript could be satisfactory for publication in  International Journal of Molecular Sciences.

The responses to the comments please see the attachment.

Reviewer 3 Report

The objective of the work as well as the novelty needs to be described in detail within the abstract.  

Include a brief summary of the major findings of the work within the abstract.

Since the topic of this work is about PIV and optical flow technique, its important that the authors include the most recent developments in this field of research.  The most recent development is 3D additive manufacturing techniques of arterial networks as well as compliant vessels as shown below:

Stanley, N., Ciero, A., Timms, W., and Hewlin, R. L., Jr.A 3-D Printed Optically Clear Rigid Diseased Carotid Bifurcation Arterial Mock Vessel Model for Particle Image Velocimetry Analysis in Pulsatile Flow ASME Open J. Engineering ASME. January 2023 2 021010 doi: https://doi.org/10.1115/1.4056639

Hewlin, R.L., Kizito, J.P. Development of an Experimental and Digital Cardiovascular Arterial Model for Transient Hemodynamic and Postural Change Studies: “A Preliminary Framework Analysis”. Cardiovasc Eng Tech 9, 1–31 (2018). https://doi.org/10.1007/s13239-017-0332-z

The introduction and background should provide bullet points of the major and even minor contributions of this work that highlights the novelty of this research.

The resolution of Figure 1 and other images need to be improved.  The scales are almost illegible.  Increase the size of figures and scales.

There is significant white space in the manuscript that also needs to be addressed.

A conclusion needs to be added that restates the ultimate goal, provides a summary of the major findings and bullet points the limitations of this work as well as highlight areas for future work.

The manuscript needs to be reviewed for grammatical errors

Author Response

(The authors gave the same response as above.)

Round 2

Reviewer 2 Report

Major changes have been made to the text according to my comments. I agree. I proposed to publish the article in this revised version. However, I would like to note for authors, that authors do not understand the clinical significance of their technique quite correctly (remark 1), in my opinion. Indeed, knowledge of the blood flow (BF) in a microvascular bed is important in clinical meaning. But this knowledge is not about the velocity of BF in a single small vessel. There are million units of such vessels in organs, and all of them cannot be examined by the proposed method. The proposed method can examine tens of vessels, no more. And this is less than 0,01%, thus it is not clinically significant. Doctors need information about the entire microcirculatory bed, not about 10 small vessels. Therefore, this technique will not give practical assistance for clinicians. This is just my comment, just for a case. 

Reviewer 3 Report

None